# A Shallow Pooled Weighted Feature Enhancement Network for Small-Sized Pine Wilt Diseased Tree Detection

Mei Yu [1,2], Sha Ye [1], Yuelin Zheng [1,2,*], Yanjing Jiang [1], Yisheng Peng [1], Yuyang Sheng [1], Chongjing Huang [1] and Hang Sun [1,2]

1 College of Computer and Information Technology, China Three Gorges University, Yichang 443002, China; yumei@ctgu.edu.cn (M.Y.); ys@ctgu.edu.cn (S.Y.); jiangyanjing@ctgu.edu.cn (Y.J.); pys@ctgu.edu.cn (Y.P.); shengyuyang@ctgu.edu.cn (Y.S.); huangchongjing@ctgu.edu.cn (C.H.); sunhang0418@whu.edu.cn (H.S.)
2 Hubei Key Laboratory of Intelligent Vision Based Monitoring for Hydroelectric Engineering, China Three Gorges University, Yichang 443002, China
* Correspondence: zhengyuelin@ctgu.edu.cn; Tel.: +86-132-942-443-30

**Abstract:** Pine wild disease poses a serious threat to the ecological environment of national forests. Combining the object detection algorithm with Unmanned Aerial Vehicles (UAV) to detect pine wild diseased trees (PWDT) is a significant step in preventing the spread of pine wild disease. To address the issue of shallow feature layers lacking the ability to fully extract features from small-sized diseased trees in existing detection algorithms, as well as the problem of a small number of small-sized diseased trees in a single image, a Shallow Pooled Weighted Feature Enhancement Network (SPW-FEN) based on Small Target Expansion (STE) has been proposed for detecting PWDT. First, a Pooled Weighted Channel Attention (PWCA) module is presented and introduced into the shallow feature layer with rich small target information to enhance the network's expressive ability regarding the characteristics of two-layer shallow feature maps. Additionally, an STE data enhancement method is introduced for small-sized targets, which effectively increases the sample size of small-sized diseased trees in a single image. The experimental results on the PWDT dataset indicate that the proposed algorithm achieved an average precision and recall of 79.1% and 86.9%, respectively. This is 3.6 and 3.8 percentage points higher, respectively, than the recognition recall and average precision of the existing state-of-the-art method Faster-RCNN, and 6.4 and 5.5 percentage points higher than those of the newly proposed YOLOv6 method.

**Keywords:** pine wilt disease; shallow feature map; channel attention; data enhancement

## 1. Introduction

Pine wild disease (PWD), known as the pine killer, poses a significant threat to pine forests globally [1]. This disease is caused by the pine wood nematode, which infiltrates and reproduces within the pine tree, ultimately resulting in the tree's demise [2].

At present, effective prevention and control measures involve manually cutting down infected pine trees affected by pine wilt disease, followed by centralized burning of the felled diseased trees. Additionally, a special medicine is sprayed on the stumps of the diseased trees and sealed to prevent secondary transmission. An important prerequisite for the above-mentioned control measures is the identification and localization of infected pine trees, which is achieved through the detection of diseased trees. Traditional monitoring of pine tree blights mainly relies on manual detection. Staff observe the appearance and surface morphological characteristics of trees, judging based on the color change characteristics of infected pine trees, such as yellowish-brown and reddish-brown [3]. This method has the disadvantages of poor timeliness and large recognition errors, making it difficult to effectively complete the task of epidemic monitoring.

Compared with manual detection, aerial remote sensing image monitoring has the advantages of wide coverage, low labor intensity, and high efficiency. However, imple-

menting satellite remote sensing image monitoring has high costs, low resolution, poor timeliness, and is easily disturbed by natural environmental factors.

With the development of UAV technology, high-resolution UAV imagery has brought great convenience to monitoring tasks in various industries [4–6]. A feasible method for early detection of infected trees is using aerial images obtained by unmanned aerial vehicles (UAVs). In recent years, deep learning technology has rapidly developed; deep learning technologies have been used in various research fields, such as image defogging [7], face recognition [8,9], video object segmentation [10], etc. Due to its powerful feature extraction capabilities, researchers have begun to apply the combination of UAV remote sensing technology and deep learning technology to various fields [11].

Xu Xinluo et al. [12] used the Faster R-CNN algorithm to automatically identify pine blight diseased trees and locate infected pine trees, achieving a recognition accuracy of 82.42%. However, this method uses a two-stage detection network, and the reasoning speed for diseased trees is slow. Additionally, the amount of data in the experiment is small, which is insufficient to be applied to the monitoring task of actual pine wilt diseased trees. Li Fengdi et al. [13] used the improved YOLOv3-CIoU algorithm to detect PWDT and improved the accuracy of the disease by citing the more accurate regression loss function CIoU. However, the research area was only 0.275 square kilometers, and the research results are not representative. Bingxi Qin et al. [14] used the improved YOLOv5 algorithm to detect multispectral data of pine wood nematodes from UAVs and achieved relatively good results with high identification accuracy. However, the acquisition efficiency of multispectral data is low, the technical requirements for UAV flights are high, and the acquisition cost of diseased tree images is high. As a result, it is not suitable for large-scale identification of pine wilt diseased trees.

Although the above studies have achieved certain results in the detection of pine wild diseased trees (PWDT), they all have the following problems: there are too few small diseased tree samples in a single picture in the training dataset, and the complex background is easy to interfere with the algorithm's detection of small diseased trees in the shallow feature map. Feature extraction leads to missed detection of a large number of small-sized diseased trees when detecting diseased trees.

To address these issues, this paper proposes a Shallow Pooled Weighted Feature Enhancement Network (SPW-FEN) for small PWD tree detection in UAV images. The proposed network takes advantage of both shallow and deep features, and applies pooling and weighting schemes to enhance the discriminative power of features. Specifically, in this paper, we propose a Shallow Pooled Weighted Feature Enhancement Network (SPW-FEN) based on Small Target Extension (STE) for PWDT detection. First, two layers of shallow feature maps are used to split the output of small-sized diseased trees. At the same time, a Pooled Weighted Channel Attention module (PWCA) is presented, which introduces the proposed PWCA module into the shallow layer of the FPN structure to enhance the feature response of the small diseased tree target in the shallow feature map and enhances the algorithm's ability to extract the features of the small-sized diseased trees.

In addition, in small-target detection, data augmentation technology can increase the number of samples in the training set by rotating, translating, scaling, etc., thereby improving the detection ability of the model. In the latest YOLO series algorithms, such as YOLOv4, YOLOv5, etc., data enhancement methods are used to improve the detection ability of the algorithm for small targets by increasing the number of small targets. However, the expanded samples from the data enhancement method used above have problems such as deformation, color gamut transformation, focusing on the expansion of the overall sample, and do not show obvious expansion of the small target sample, which is not applicable to the small-scale pine wilt diseased trees in this paper. Based on this, we propose an STE data enhancement method. While increasing the sample size of small and medium diseased trees in a single image, the robustness of the algorithm is improved.

The proposed network is evaluated on the pine wilt diseased trees dataset containing UAV images of PWDT. Experimental results show that SPW-FEN outperforms several

state-of-the-art methods in terms of detection accuracy, especially for small PWDT. In addition, a comprehensive analysis is performed to study the effectiveness of the proposed pooling and weighting schemes, as well as the contribution of shallow and deep features.

The remaining chapters of this paper are arranged as follows: Section 2 introduces in detail the dataset of pine wilt diseased trees produced in this paper, the experimental environment used in this paper, the design of experimental parameters, and a detailed description of the proposed SPW-FEN method. In Section 3, the results of our comparative experiments and ablation experiments are summarized and analyzed. Finally, Section 4 concludes the paper and discusses future directions.

## 2. Materials and Methods

### 2.1. UAV Pine Forest Image Acquisition

UAVs equipped with high-resolution cameras were used to take images of pine forests in Yiling District and Yidu City of Yichang City according to fixed routes. Among them, the UAV model was MD-25 UAV. This model is powered by four T-MOTOR motors and TMOTOR flame high-voltage electronic governors to provide rotor power; one T-MOTOR motor is matched with T-MOTOR high-voltage, and the electronic governor provides fixed-wing power. Power device type: electric brushless engine, electronic speed control system; control device type: micro servo steering gear. The overall appearance of the MD-25 UAV is shown in Figure 1, and the main parameters of the MD-25 drone casing are shown in Table 1 below.

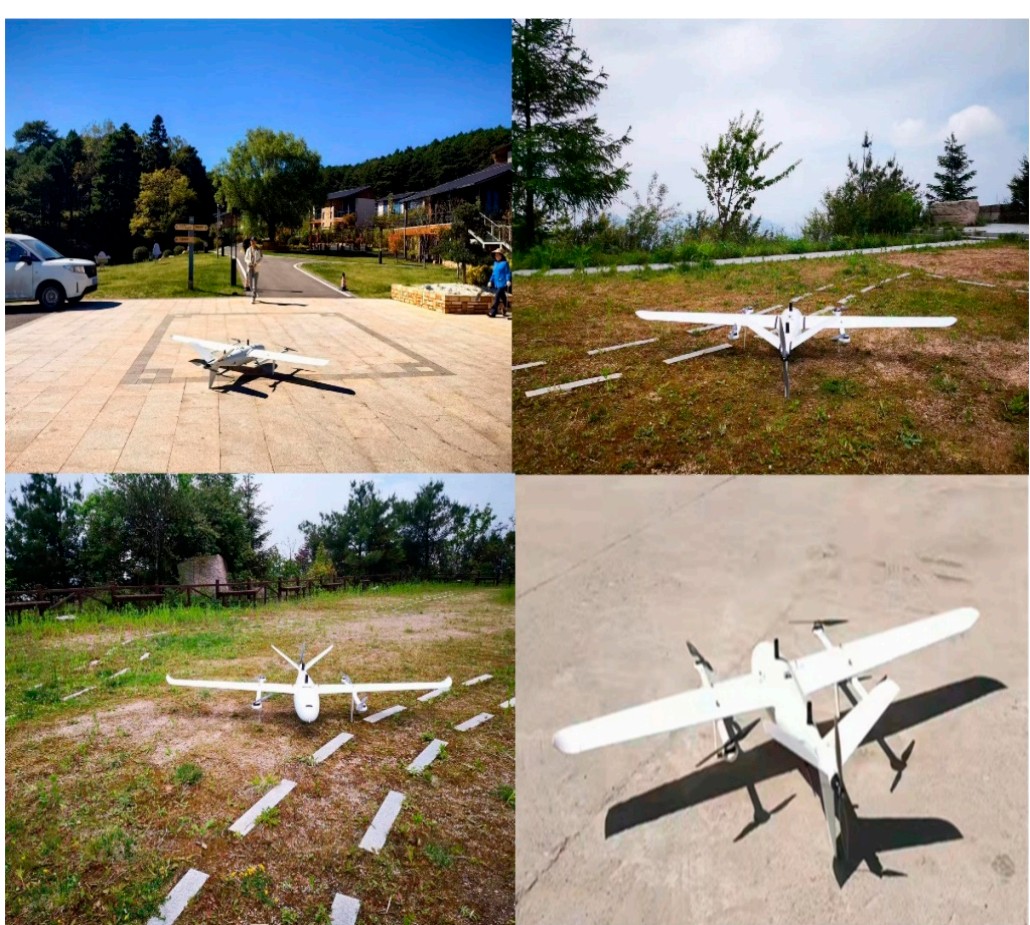

**Figure 1.** MD-25 drone.

**Table 1.** The main parameters of the MD-25 drone.

| Parameters | Attribute |
|---|---|
| Body material | carbon fiber, glass fiber, Kevlar, PVC, etc. |
| Maximum take-off weight | 13.5 kg |
| Maximum payload | 3 kg (standard load: 1.2 kg) |
| Wing area | about 52 dm$^2$ |
| Wing load | about 240 g/dm$^2$ @ 12.5 kg |
| Standard cruising speed | 19 m/s @ 12.5 kg |
| Maximum cruising speed | 93.6 km/h |
| Standard battery configuration | 45.6 V |
| Stall speed | 15.5 m/s @ 12.5 kg |
| Minimum circling radius | 120 m @ 19 m/s |
| Fixed-wing maximum thrust-to-weight ratio | 0.6 |

The cameras were Zeiss 35 mm fixed-focus lens, 36 million pixels, as shown in Figure 2.

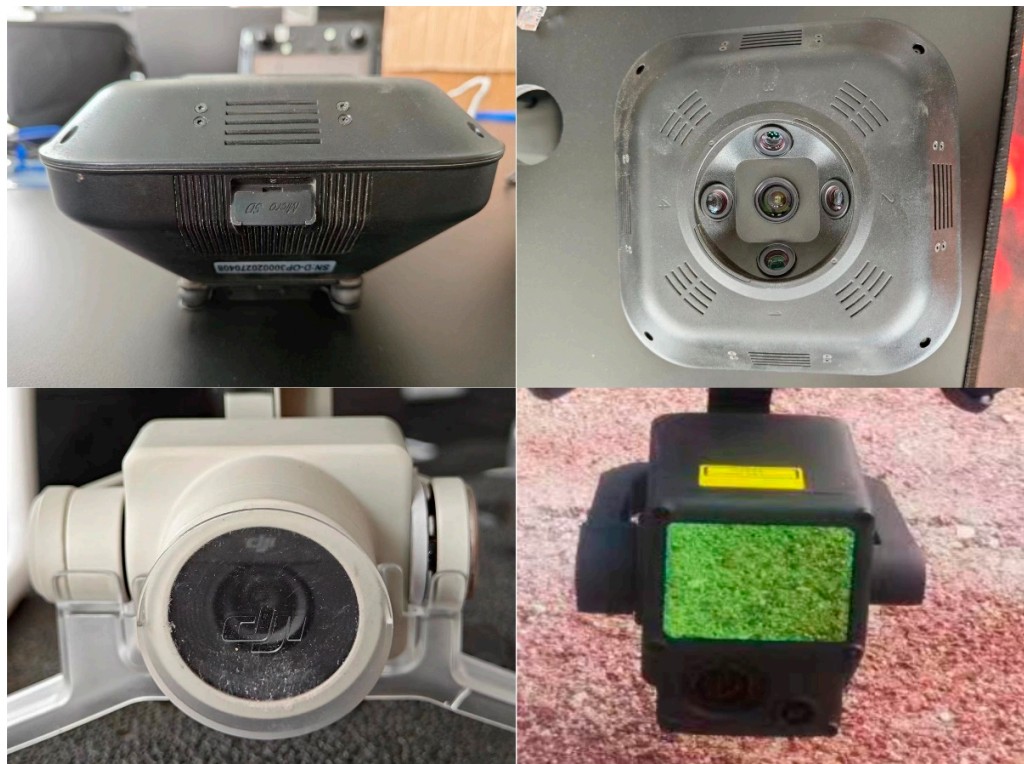

**Figure 2.** High-resolution cameras onboard drones.

In terms of route setting: set the relative flight height to 350 m, the average ground resolution to 4.89 cm, the flight height difference between adjacent photos on the same route to ≤30 m, and the difference between the actual flight height and the design flight height to ≤50 m; the heading overlap is 70%, the lateral overlap is 35%, and the single flight is 70 km. The sun elevation angle at the time of photography is greater than 30–40°.

*2.2. Pine Wilt Diseased Tree Dataset*

The pine forest images taken according to the UAV, on-board camera, and route in the previous section were used as the data source of the dataset. Crop the obtained drone image with a pixel size of 7952 × 5304 to a size of 1000 × 1000 pixels, as shown in Figure 3:

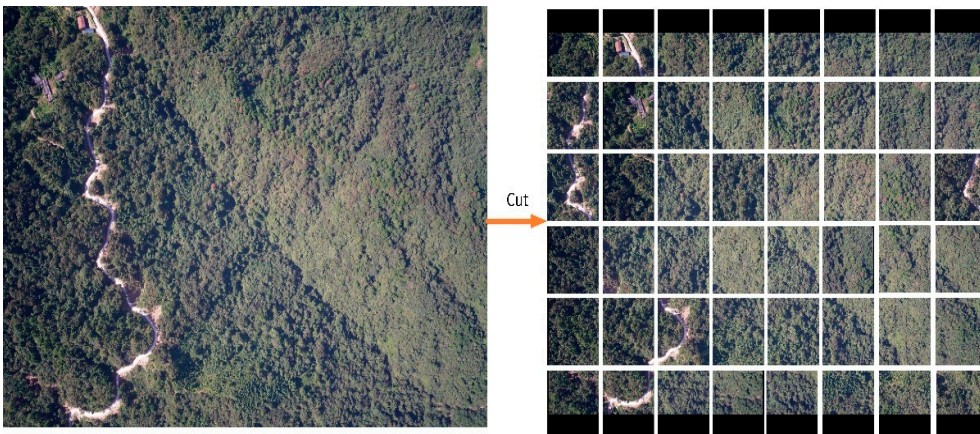

**Figure 3.** Cropping of drone images.

Then, use the LabelImg tool to label the pine wilt diseased trees in the cropped 1000 × 1000-pixel image, in which the red non-diseased trees, yellow bare land, and red roofs that are prone to interference are marked as negative sample classes, as shown in Figure 4.

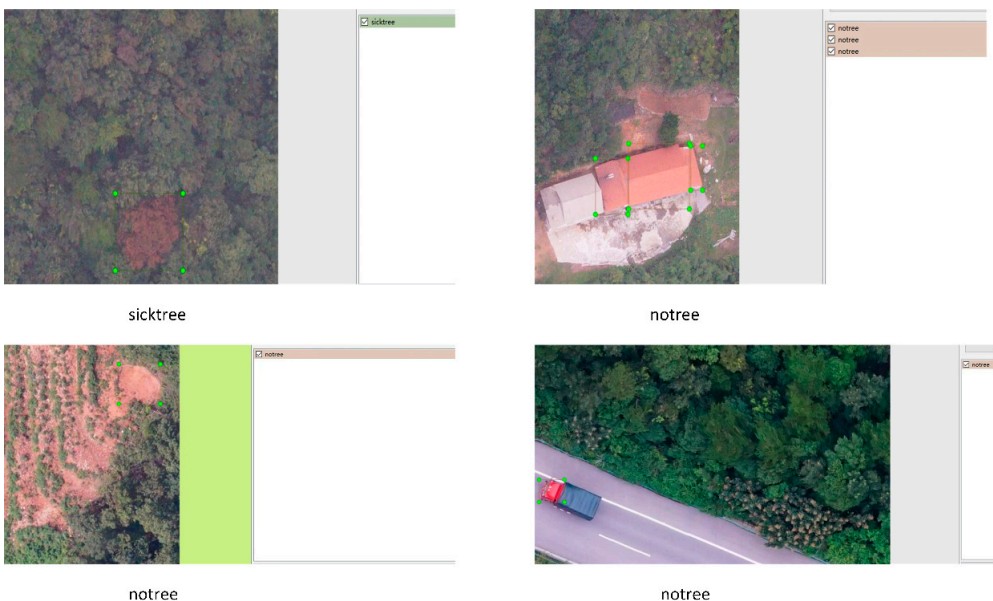

**Figure 4.** Annotated map of positive and negative samples. Mark the red disease-like trees that are easy to interfere with the detection, the red bare land, and the red house as a category, and name it notree.

A total of 3271 positive samples of diseased pine trees and 1000 negative samples of easily disturbed diseased trees were marked, and then the marked image data were divided into training datasets, validation datasets, and test datasets. Among them, according to the pixel size of the diseased tree, the classification method of the COCO dataset defines targets with a diseased tree target pixel area smaller than 32 × 32 pixels as a small target, targets with an area between 32 × 32 and 96 × 96 pixels as a medium object, and objects whose area is larger than 96 × 96 is defined as a large object [15]. See Table 2 for more information on the dataset.

**Table 2.** Classification of the PWDT dataset.

| Dataset | Pictures (Sicktree) | Pictures (Notree) | Small Targets | Medium Targets | Large Targets |
|---|---|---|---|---|---|
| Training Set | 2648 | 1000 | 727 | 4239 | 521 |
| Validation Set | 295 | 0 | 64 | 456 | 50 |
| Test Set | 328 | 0 | 68 | 538 | 63 |

It can be seen from Table 2 that in our PWDT dataset, the medium-sized diseased trees accounted for the largest proportion, and the number of small-sized diseased trees and large-scale diseased trees was relatively small. The number of small target diseased trees in the training set was 727, accounting for 13.2% of the total target number. The number of small target diseased trees in the validation dataset and the test dataset was relatively small.

Figure 5 shows in detail the proportion of small target diseased trees, medium target diseased trees, and large target diseased trees in the training set and verification set. In the training set, small target diseased trees accounted for 13% and medium target diseased trees and large target diseased trees accounted for 77% and 10%, respectively, while in the verification set, small target diseased trees accounted for 11% and medium target diseased trees and large target diseased trees accounted for 80% and 9%, respectively. In the pine wilt diseased tree dataset produced in this paper, the medium target diseased trees accounted for the vast majority, and the small target diseased trees and large target diseased trees accounted for a small proportion, resulting in uneven distribution of diseased trees within the class.

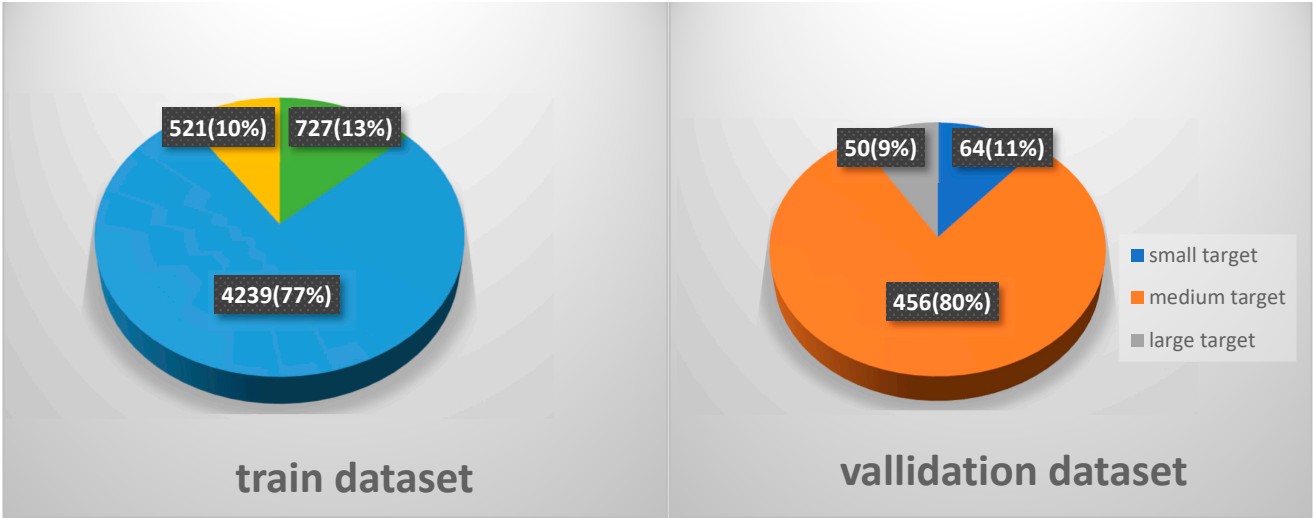

**Figure 5.** The proportion of each target in the training dataset and validation dataset.

*2.3. Method*

In this section, we first describe the Shallow Pooled Weighted Feature Enhancement Network (SPW-FEN) in Section 2.3.1, and then present the Pooled Weighted Channel Attention (PWCA) module in Section 2.3.2. Finally, Section 2.3.3 illustrates the proposed STE data enhancement method.

2.3.1. Shallow Pooled Weighted Feature Enhancement Network (SPW-FEN)

The proposed SPW-FEN uses a ResNet50 [16] feature extraction network to extract features to generate feature maps C1, C2, C3, C4, and C5. At the neck of the network, the feature pyramid structure is used to fuse the shallow feature map with high resolution and the deep feature map with low resolution and rich semantic information. Among them, the shallow feature map has more edge position information about the small-sized target,

which is conducive to the detection of small-sized PWDT; the deep feature map has more semantic information, while the small target occupies fewer pixels in the image. In the deep feature map after multi-layer convolution, the feature information is easy to be lost.

The RetinaNet [17] algorithm uses the prediction feature map P3 that combines the feature map C4 and the shallow feature map C3 to predict the output of small-sized targets. The feature maps C3 and C4 have lost some details for some small-sized targets.

The proposed algorithm adds the prediction feature map P2, which combines the shallow feature maps C3 and C2 to divide the small-sized diseased tree targets into small-sized targets and minimum-sized targets, respectively, in the prediction feature map P3, and the P2 layer carries out shunt prediction output. At the same time, we introduce the PWCA module behind the shallow feature maps C2 and C3 to enhance the feature response ability of the shallow feature layer to small-sized targets.

In addition, based on the statistics of the scale distribution of diseased tree targets in the pine wilt diseased tree dataset, it was found that the minimum-scale diseased trees with target scales less than $24 \times 24$ pixels in the PWDT dataset accounted for 4.4%, and the number was 295; the proportion of diseased trees with a target scale greater than $256 \times 256$ pixels in the dataset was 0%. Therefore, the proposed network model adds the shallow feature map P2, deletes the deep feature maps P6 and P7, and designs the anchor frame size of each layer. According to the distribution of target scales in the dataset, set the anchor size on the prediction feature maps P2 to P5 to $16 \times 16$, $36 \times 36$, $78 \times 78$, and $140 \times 140$, respectively. Figure 6 is the structure of the SPW-FEN network proposed in this paper.

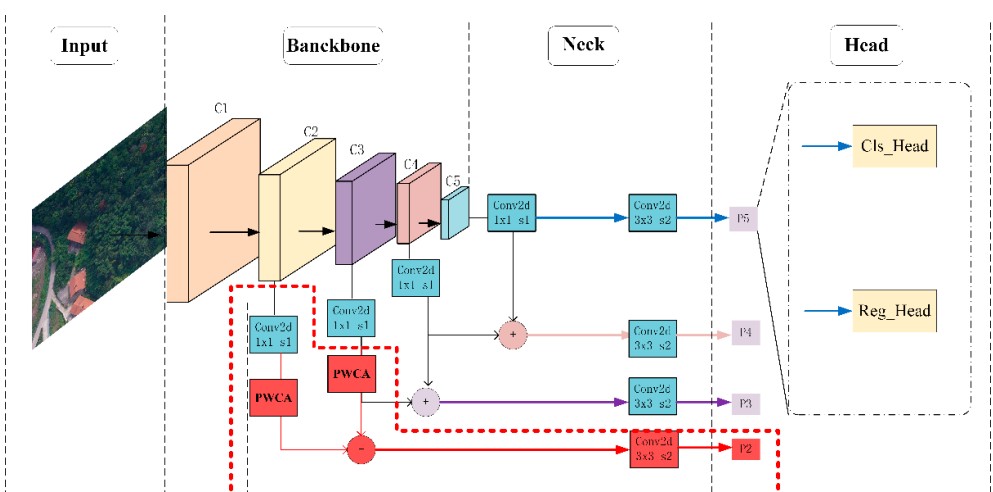

**Figure 6.** Structure of the Shallow Pooled Weighted Feature Enhancement Network. The proposed network uses resnet50 to extract features and then inputs the feature map into the feature fusion module. The pooled weighted attention module is introduced before the feature fusion of the shallow feature maps C2 and C3 to enhance the network's ability to express the features of small-sized targets.

### 2.3.2. Pooled Weighted Channel Attention (PWCA) Module

A large number of research results show that the channel attention module is conducive to the feature extraction of the target area by the network and can effectively mitigate the effect of background information on the feature extraction of small-sized targets [18–21]. To enhance the feature extraction ability of the shallow feature layer for small-scale diseased trees, in this paper, we propose a PWCA module, which is added after the $1 \times 1$ convolution operation of the shallow feature images C2 and C3. The PWCA module can increase the attention weight of the network model to the diseased tree area, inhibit the characteristic response of the background area, increase the network model's ability to distinguish small target feature channels and background channels and improve the network's detection performance of small-scale diseased trees. The structure of the PWCA module is shown in Figure 7.

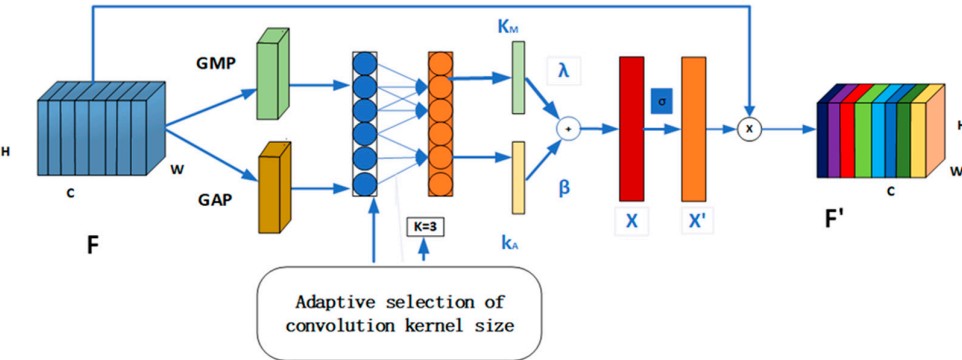

**Figure 7.** Pooled Weighted Channel Attention Module (PWCA). The input feature map is pooled using global average pooling and global maximum pooling, and then two one-dimensional weight parameters are obtained via a one-dimensional convolution operation. The two one-dimensional weight parameters are added using weighted fusion to obtain the attention weight value of the pooled weighted channel. The weight value is the point multiplied by the input feature map to obtain the final output feature map.

First, the global average pooling (GAP) and global maximum pooling (GMP) operations were performed on the feature graph F with dimensions H × W × C output from the backbone network to obtain two one-dimensional feature vectors of 1 × 1 × C with different spatial context information; then, the two one-dimensional eigenvectors of 1 × 1 × C obtained using GAP and GMP were, respectively, convolutional to generate two sets of channel weight values, where $K$ was adaptively determined by the mapping of channel dimension $C$, as shown in Formula (1):

$$k = \phi(C) = |\frac{log_2(C)}{\gamma} + \frac{b}{\gamma}|_{odd} \tag{1}$$

where $\gamma = 2$, $b = 1$, and $K$ is the odd number of the neighboring calculation.

The $K_A$ weights of the two channels are adaptively added $K_M$. Additionally, they are fused according to the random weighting to obtain the pooled weighted attention channel weights $X$, as shown in Formula (2):

$$X = \lambda K_M + \beta K_A \tag{2}$$

where $\lambda$ and $\beta$ are two super parameters.

Additionally, the weight is then normalized to 0–1 through the sigmoid activation function to obtain the attention weight. The $X'$ obtained attention weight is dot multiplied $X'$ with the original feature map F to obtain the attention feature map $F'$, as shown in Formula (3):

$$F\prime = F \times X\prime \tag{3}$$

### 2.3.3. Small Target Expansion (STE) Data Enhancement Method

The total number of small-sized diseased trees in our PWDT dataset and the number of small-sized diseased trees in a single image is small, and a small number of small-sized diseased tree data is not enough for the feature extraction network to extract their features. Therefore, in this paper, we propose an STE data enhancement method based on small-sized targets with double fixed scaling. First, through the fixed scale scaling method, four pictures numbered 1, 2, 3, and 4 are randomly selected from the pine wilt diseased tree dataset, and the length and width of the four pictures are scaled to the same ratio of 0.4, 0.5, and 0.6 to obtain the scaled picture $Img1$, $Img2$, $Img3$, and $Img4$, as shown in Formula (4):

$$Img_i = resize(random(0.4, 0.5, 0.6)jpg_i)(i \in [1, 4]) \tag{4}$$

Next, create a new rectangular box whose length and width are twice the size of the picture in the pine wilt diseased tree dataset. Take the center of the square box as the dividing point, and divide the rectangular box into four sub-areas, r1, r2, r3, and r4, of the same size. Then, fill the pictures *Img*1, *Img*2, *Img*3, and *Img*4 randomly into the sub-regions r1, r2, r3, and r4, reduce the length and width of the filled rectangular box by two, and the resulting rectangular box is an expanded sample image. Finally, remove the scaled and spliced pictures from the pine wilt tree dataset and repeat the above steps in the remaining pictures; a large number of expanded sample graphs were obtained and stored in the PWDT dataset, and the specific operation flow is shown in Figure 8.

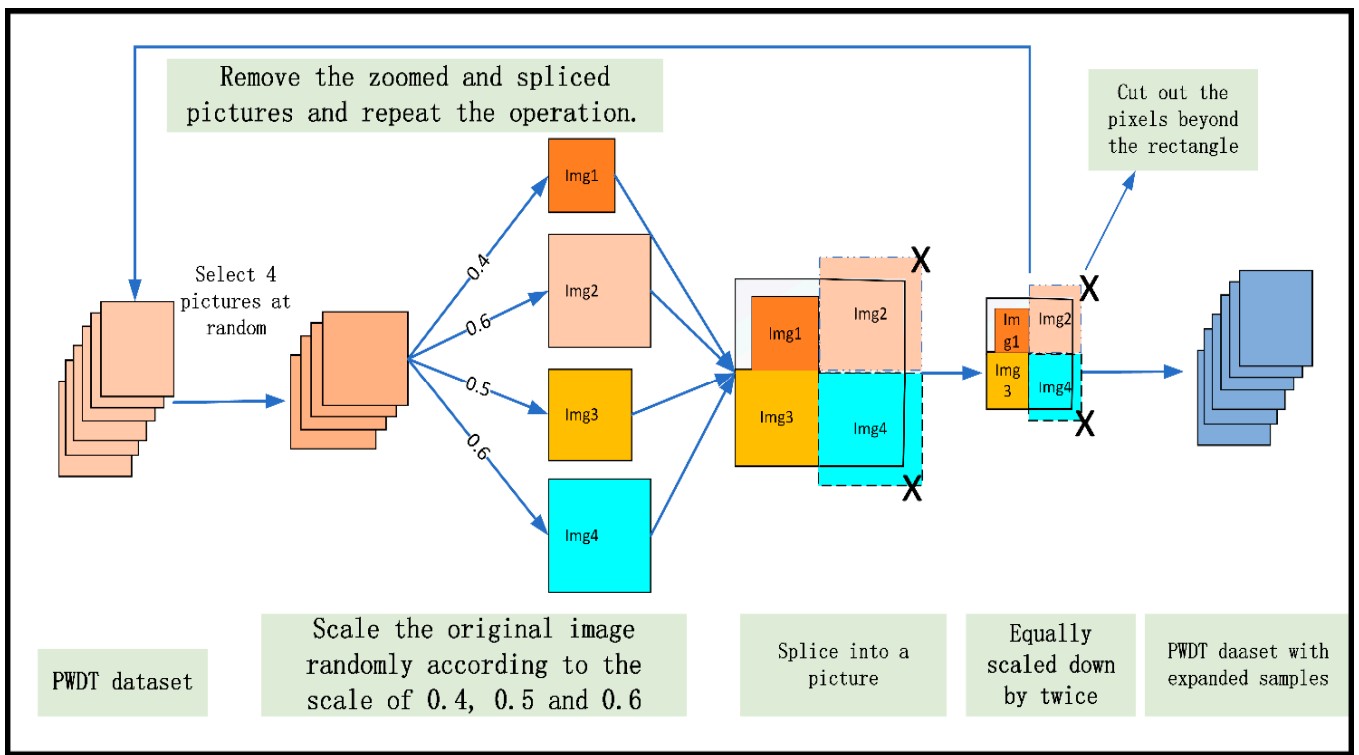

**Figure 8.** Flow chart of STE data enhancement. Four pictures are randomly selected from the dataset, and the expanded samples with rich small-sized targets are obtained through splicing and two fixed-scale scaling.

## 3. Results

In this section, we first introduce our experimental environment. Then, we introduce the evaluation metric of our experimental results, and then compare our algorithm with several current mainstream object detection algorithms on our dataset. Finally, an ablation experiment is designed for the proposed modules.

### 3.1. Experimental Environment and Parameter Setting

The detection algorithm in this paper is based on the PyTorch framework and uses NVIDIA GeForce RTX 3090. Using the dataset of PWDT made by ourselves to train the network model, a total of 120 epochs were trained in this experiment, and the learning rate was adjusted at the 80th and 110th epochs. The initial learning rate was set to 0.0001, and the batch size was set to four. The experimental environment and experimental parameter settings are shown in Table 3.

**Table 3.** Experimental environment and experimental parameter settings.

| Name | Version Number/Parameter |
|---|---|
| GPU | Nvidia GeForce GTX 3090 GPU |
| Server memory | 64 G |
| Operating system | Ubuntu 18.04 |
| Deep learning framework | Pytorch 1.8.0 |
| Epoch | 120 |
| Initial learning rate | 0.0001 |
| Batch-size | 4 |
| Momentum setting | 0.9 |
| Regularization coefficient | 0.0001 |

*3.2. Evaluation Metric*

Target detection algorithm evaluation indicators are mainly divided into two categories: classification indicators and localization indicators.

Classification indicators: These mainly measure the classification ability of the algorithm for the target category. Commonly used indicators are *Accuracy*, *Precision*, *Recall*, and *F1-score*. Among them, the accuracy rate is an indicator to measure the overall classification of the algorithm, while the precision rate and recall rate pay more attention to the classification of a single target category by the algorithm. *F1-score* is a comprehensive index of precision rate and recall rate, which can more comprehensively evaluate the classification ability of the algorithm. It is defined as the harmonic mean of precision rate and recall rate. Its formula is as follows:

$$F1 - score = \frac{2}{\frac{1}{precision} + \frac{1}{recall}} = 2 \times \frac{precision \times recall}{precision + recall} \tag{5}$$

Positioning index: It mainly measures the evaluation of the algorithm on the target positioning ability. Commonly used indicators are the Intersection over Union (IoU), average precision (AP), and mean average precision (map). The IoU is an indicator for measuring the accuracy of the algorithm for target positioning; AP average accuracy is one of the indicators for evaluating image retrieval results. It is the abbreviation of average precision, which means that for a set of query images, all the prediction results are averaged. AP is calculated by sorting the retrieval results and calculating the area of recall and precision. For each query image, by comparing the similarity between the predicted result and the ground truth label, a set of ranked lists can be generated where each retrieved result has a relevance score. Sort these scores from high to low, and calculate the precision at each recall. Finally, the AP can be obtained by taking the average of the accuracy rates under all recall rates, and the formula is as follows; mAP considers the classification and positioning capabilities of the algorithm for all target categories, and the calculation formula of AP is as follows:

$$P = \frac{T_P}{T_P + F_P} \tag{6}$$

$$R = \frac{T_P}{T_P + F_N} \tag{7}$$

$$AP = \int_0^1 P(R)dR \tag{8}$$

where $T_P$ represents the number of samples with actual positive labels that are correctly classified as positive. $F_P$ indicates the number of samples with actual negative labels that are incorrectly classified as positive. $F_N$ denotes the number of samples with actual positive labels that are incorrectly classified as negative. $P$ represents precision, and $R$ represents recall.

In practical scenarios, object detection algorithms are evaluated based on both classification and localization indicators to comprehensively assess their performance. However, for specific applications, different indicators may need to be selected based on the specific conditions and requirements.

In the case of detecting pine wilt diseased trees, the priority is to minimize missed detections to prevent the spread of the disease. Hence, this study uses recall rate and average precision as performance indicators, where the recall rate measures the proportion of predicted positives to all annotated positives. It is expected that the model's recall rate is as high as possible while ensuring a high overall performance AP.

### 3.3. Comparative Experimental Results

To verify the performance of our proposed network model, we compared the verification results of the current seven mainstream target detection algorithms and our proposed detection algorithms on the PWDT dataset through experiments; the experimental results can be seen in Table 4 below.

**Table 4.** Experimental results of different methods on the PWDT dataset.

| Network Model (Year) | Basic Network | Recall | AP |
|---|---|---|---|
| Faster-RCNN (NeurIPS2015) | ResNet50 | 83.3 | 75.3 |
| SSD (ECCV2016) | VGG16 | 80.4 | 73.7 |
| YOLOv3 (CVPR2018) | DarkNet53 | 72.9 | 70.1 |
| FoveaBox (TIP2020) | ResNet50 | 82.4 | 77.2 |
| ATSS (CVPR2020) | ResNet50 | 80.2 | 78.3 |
| YOLOF (CVPR2021) | ResNet50 | 81.5 | 78.0 |
| YOLOv6 (arXiv2022) | EfficientRep | 80.5 | 73.6 |
| Ours | ResNet50 | 86.9 | 79.1 |

The experimental results show that compared with the classic network Faster-RCNN [22] and the mainstream network SSD [23], YOLOv3 [24], ATSS [25], YOLOF [26], FoveaBox [27], and YOLOv6 [28], the proposed detection algorithm achieves the best detection results, with a recall and AP of 86.9 and 79.1, respectively. The visual identification comparison results of each network on the test set are shown in Figure 9.

It can be found from the experimental comparison results in the two test samples in Figure 9 that the SPW-FEN algorithm proposed in this paper has the best recognition effect in small-sized pine wilt diseased trees. YOLOv3, Faster-RCNN, and ATSS all have obvious missed detections. The method proposed in this paper has greatly alleviated the missed detection of small-sized diseased trees, and the recognition effect is the best.

### 3.4. Ablation Study

To further analyze the impact of the proposed channel attention module and data enhancement module of this paper on the network performance, we used RetinaNet as the base network, and the effectiveness of the designed method will be discussed in the following three aspects: small-sized diseased tree shunt prediction output, anchor box recalibration, and PWCA module. The specific experimental analysis data are shown below.

#### 3.4.1. Small-Scale Diseased Tree Shunt Prediction Output

In order to verify the effectiveness of the small-sized disease tree shunt prediction output proposed in this paper, a comparative experiment was designed to analyze the results of only the P3 layer predicting output for small-scale diseased trees and using both the P2 layer and P3 layer to predict small-scale diseased tree output. The detection effect and the specific experimental data are shown in Table 5 below.

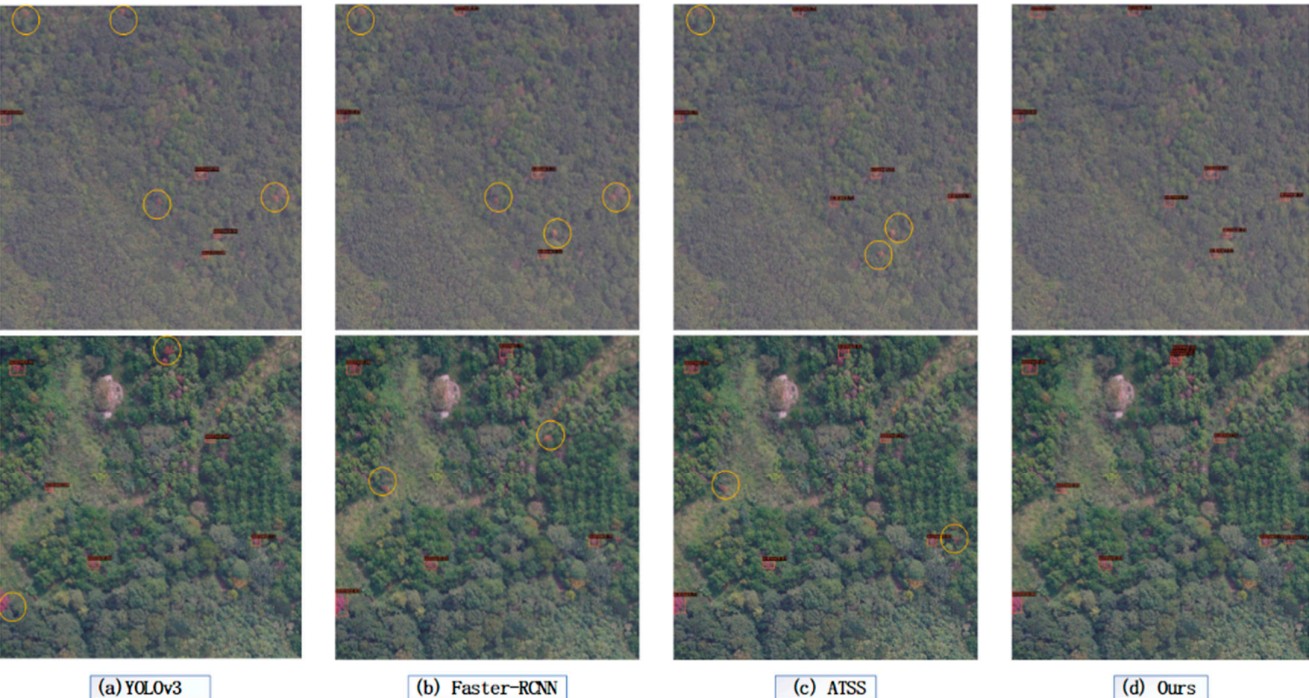

**Figure 9.** The recognition results of different methods on small-sized diseased trees in two test sample images. (**a**) shows the recognition result of YOLOv3. (**b**) shows the recognition result of Faster-RCNN. (**c**) shows the recognition result of the anchor-free detection algorithm ATSS. Additionally, (**d**) shows the recognition result of the method proposed in this paper. The red box in the figure represents the detected diseased tree, and the yellow circle represents the missed diseased tree.

**Table 5.** Comparison of recognition AP and Recall of small-size disease tree shunting prediction output module ablation experiment.

| Experiment | P2 | P3 | Recall | AP |
|---|---|---|---|---|
| Original settings (base network) | | √ | 82.4 | 77.1 |
| Split prediction output | √ | √ | 83.2 | 78.0 |

It can be seen from Table 5 that when only the P3 layer prediction feature map is used to predict the small-scale diseased tree output, the recall rate is 82.1, and the precision is only 77.1. When the P2 layer prediction feature map and the P3 layer prediction feature map are used at the same time when the scale disease tree is used for prediction output, the recall rate is increased by 1.2 percentage points, and the precision is increased by 0.9 percentage points. The recall rate and precision reach between 83.2 and 78.0, respectively. It can be seen that it is necessary to split the diseased tree for prediction output.

### 3.4.2. Recalibration of Anchor Boxes

According to the distribution of target scales in the dataset, set the sizes of the anchors on the prediction feature maps P2 to P5 to $16 \times 16$, $36 \times 36$, $78 \times 78$, and $140 \times 140$, respectively, and the three aspect ratios of the anchors to, respectively $\{1.0; 2.0; 0.5\}$ and the ratio of the area of the anchor to $\{2^0, 2^{1/3}, 2^{2/3}\}$. According to the size, aspect ratio, and the area of the anchor box, nine kinds of anchors are redesigned at each pixel on each layer of prediction feature layer. The comparison between the size of the anchor box in the original algorithm and the size of the anchor box after recalibration is shown in Table 6 below.

**Table 6.** Comparison of detection effects of diseased trees with different anchor sizes.

| Experiment | P2 | P3 | P4 | P5 | P6 | P7 | Recall | AP |
|---|---|---|---|---|---|---|---|---|
| 0 (Base network) | | 32 | 64 | 128 | 256 | 512 | 82.4 | 77.1 |
| 1 | 8 | 32 | 64 | 128 | 256 | 512 | 83.4 | 77.4 |
| 2 | 12 | 32 | 64 | 128 | 256 | 512 | 83.6 | 77.8 |
| 3 | 16 | 32 | 64 | 128 | 256 | 512 | 83.2 | 78.0 |
| 4 | 8 | 36 | 78 | 140 | | | 85.2 | 77.5 |
| 5 | 12 | 36 | 78 | 140 | | | 85.1 | 78.1 |
| 6 (Ours) | 16 | 36 | 78 | 140 | | | 85.4 | 78.4 |

From the data in Table 6, it can be seen that the adjustment of the anchor size can effectively change the detection effect of the diseased tree. There is no P2 layer in the original RetinaNet [17] network, and the detection accuracy and recall rate of the diseased trees are low. When adding the P2 layer and adjusting the size of the anchor in the P2 layer when detecting the diseased tree, the precision and recall rate are significantly improved. When the anchor of the P2 layer is set to $16 \times 16$, the anchor of the P3 layer is set to $36 \times 36$, the anchor of the P4 layer is set to $78 \times 78$, and the anchor of the P4 layer is set to $140 \times 140$, the recall rate and precision, respectively, reach 85.4 and 78.4, compared with when no adjustment is made to the size of the anchor, the recall rate and precision increased by 3 percentage points and 1.3 percentage points, respectively.

3.4.3. Pooled Weighted Channel Attenuation (PWCA) Module

To validate the effectiveness of the proposed Pooled Weighted Attention (PWCA) module in this chapter, this section investigates the influence of global average pooling and global maximum pooling on the detection of pine wilt disease in trees by adjusting the weighted parameter values ($\lambda$, $\beta$). Additionally, the impact of dimensionality reduction (MLP network) on the performance of the attention mechanism is analyzed through experiments. The specific experimental data are presented in Table 7.

**Table 7.** Experimental Results of Attention Module Ablation in Pine Wilt Diseased Tree Dataset.

| Experiment | $\lambda$ | $\beta$ | MLP (Dimension Compression) | Conv $1 \times 1$ (1D Convolution) | AP |
|---|---|---|---|---|---|
| Base network | | | | | 77.1 |
| 0 (ECA) | 0 | 1 | | √ | 77.8 |
| 1 | 0 | 1 | √ | | 77.3 |
| 2 | 1 | 0 | | √ | 77.9 |
| 3 (CBAM) | 1 | 1 | √ | | 77.4 |
| 4 | 1 | 1 | | √ | 77.6 |
| 5 | 0.5 | 0.5 | | √ | 77.5 |
| 6 | 0.5 | 1.5 | | √ | 77.3 |
| 7 (PWCA) | 1.5 | 0.5 | | √ | 78.2 |

From the experimental results presented in Table 7, it can be observed that when $\lambda = 0$ and $\beta = 1$, the attention mechanism is referred to as ECA [29]. Additionally, when utilizing one-dimensional convolution instead of the dimensionality compression operation of the MLP network, the accuracy improves by 0.7% compared with that of the baseline. In this case, when the dimensionality compression operation of the MLP network is employed, the attention mechanism becomes CBAM [30]. Substituting the MLP network in the CBAM attention mechanism with one-dimensional convolution leads to a 0.5% increase in accuracy compared with the baseline. By adjusting the parameter values of $\lambda$ and $\beta$ and analyzing the weighted parameter experimental data, it is found that when $\lambda = 1.5$ and $\beta = 0.5$, the introduction of the attention mechanism has the highest recognition accuracy for the diseased tree. It is evident that the pooling weighted channel attention (PWCA) achieves the highest experimental accuracy, yielding the best detection results for diseased trees. The

experimental results on the pine wilt diseased tree dataset indicate that the MLP network has a detrimental effect on the channel attention mechanism. It proves to be inefficient and unnecessary for capturing dependencies among all channels. Conversely, considering the recognition results for pine wilt diseased trees with fewer targets in a single image, the PWCA attention mechanism with an increased weight on global maximum pooling performs better in terms of diseased tree recognition.

### 3.4.4. Comprehensive Experimental Analysis

To further analyze the impact of the proposed channel attention module and data enhancement module of this paper on the network performance, we designed the ablation experiment after adding each module on the basis of the RetinaNet algorithm. The results of the ablation experiments are shown in Table 8 below.

**Table 8.** Ablation study of every module.

| Number | Anchor Setting | PWCA | STE | Recall | AP |
|--------|----------------|------|-----|--------|------|
| 1 | | | | 82.4 | 77.1 |
| 2 | √ | | | 85.4 | 78.4 |
| 3 | | √ | | 84.3 | 78.2 |
| 4 | √ | √ | | 86.0 | 78.8 |
| 5 | | | √ | 85.5 | 77.7 |
| 6 (ours) | √ | √ | √ | 86.9 | 79.1 |

It can be seen from Table 8 that the recall of the proposed module increased from 82.4 to 85.4, the recall increased by 3, the AP increased from 77.1 to 78.4, and the AP increased by 1.3 after the anchor re-setting and the prediction output of the diversion in the RetinaNet network. After adding PWCA to the shallow feature map of the RetinaNet algorithm, the recall increased by 1.9 and the AP improved by 1.1. In the RetinaNet algorithm, the recall and AP of the algorithm were improved by 3.1 and 0.6, respectively, after the STE data enhancement method was adopted. At the same time, after using the PWCA module and STE data enhancement in the RetinaNet network, the recall was improved by 4.5 and the AP was improved by 2.0.

As shown in Figure 10a, the picture is overexposed, and light photography is brighter than natural light. As shown in Figure 10b–d, the original mosaic enhanced picture has lost the red and yellow-brown color characteristics of PWDT. Through experiments, it is found that these low-quality samples are mainly generated during HSV transformation of the image during sample enhancement [31]. To solve this problem, the STE data enhancement proposed in this paper removes HSV transformation operation, significantly improving the quality of the enhanced samples. The red circle represents the small-sized diseased tree after using the STE data enhancement method, as shown in Figure 10. Under the condition of ensuring the same quality as the original sample, the setting of the fixed scaling scale significantly increases the number of small target samples in the enhanced sample. There is only one small target or even no small target samples in the original image, and the number of small target samples in the transformed sample is increased by more than four to eleven samples, which effectively alleviates the problem of too few positive samples in the training process.

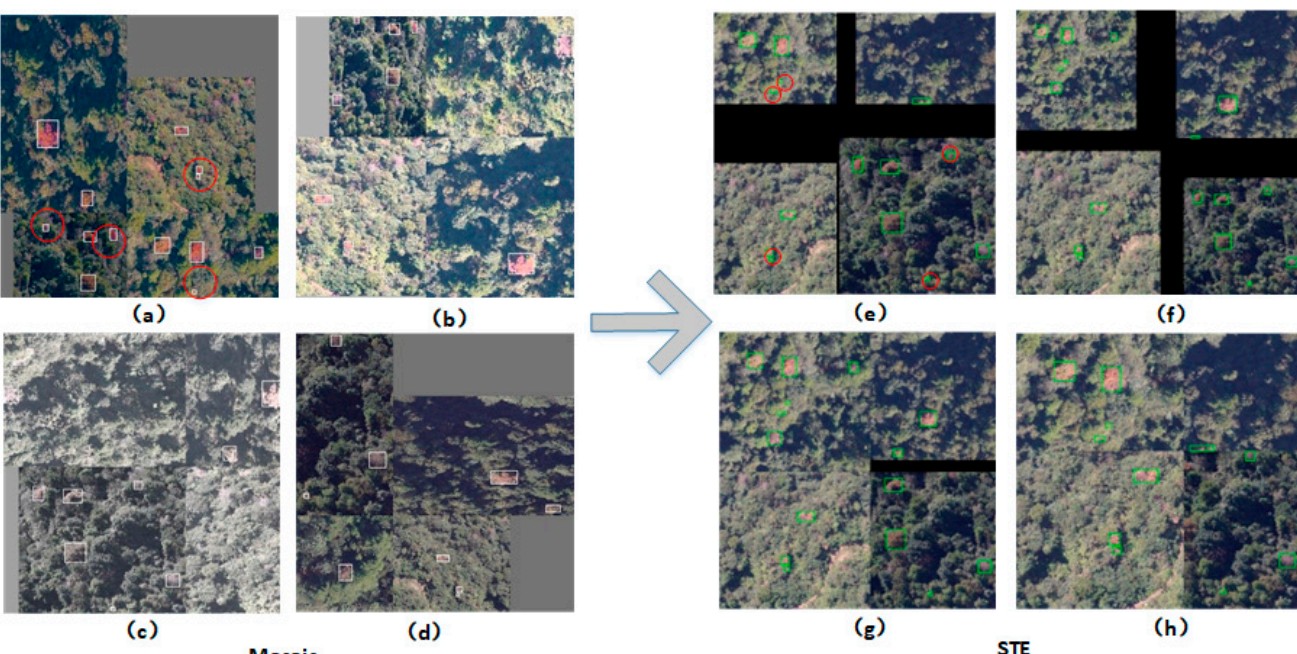

**Figure 10.** (**a–d**) represent the four pictures randomly generated in the Mosaic data enhancement method that deviate from the target color characteristics of the diseased tree, and (**e–h**) represent the four pictures generated by using the STE data enhancement method. Among them, the white box represents the labeling target in the mosaic enhancement method, the green box represents the labeling target in the expanded sample in the STE method, and the red circle represents the small-scale diseased tree. From the comparison chart, it can be seen that the STE data enhancement method has expanded the number of small-size diseased trees more.

## 4. Discussion

Through the analysis and statistics of the scale size of the diseased trees in the pine wilt diseased tree dataset, we found that the number of small-scale diseased trees is small, which is not enough for the network model to learn the characteristics of small-scale diseased trees. At the same time, we found that in drone footage, the small-scale diseased tree only occupies a small part of the pixel area in the image, and most of the pixel areas are background pixels. This background information seriously interferes with the feature extraction of the small-scale diseased tree.

As for the problem of background information interference, more and more researchers have begun to use the attention mechanism to alleviate the interference problem [32,33]. Therefore, in this paper, we propose a Pooled Weighted Channel Attention module to alleviate the background interference on small-scale diseased tree feature extraction. From the bias of the importance of global maximum pooling and global average pooling to feature learning after conducting research, a large number of experiments have proved that for the detection of small-scale diseased trees, the contribution of global maximum pooling is higher than that of global average pooling. Through the weighted fusion of global large pooling and global average pooling, exploring weight parameters is most suitable for small-scale diseased tree detection.

On the other hand, through the analysis of the advantages and disadvantages of the existing data enhancement methods in small-scale diseased tree data enhancement, we propose a data enhancement method based on small target sample expansion, so that it does not affect the color and shape of diseased tree targets. Based on the characteristics, the number of small-scale diseased trees is expanded. The experimental results show that the data enhancement method proposed in this paper can significantly enhance the number of small-scale diseased trees and the robustness of diseased tree detection.

The current research methods have achieved good results in the detection of small-scale diseased trees, but the detection effect on late-stage diseased trees is not good, and further research and analysis are needed. On the other hand, due to the high cost of acquiring diseased tree datasets, which require huge manpower and material resources, the number of existing pine wilt diseased tree datasets is relatively small. How to learn the characteristics of the target with a small number of labels is the focus of future research. At present, active learning technology is developing rapidly in various fields [34,35], and active learning mainly focuses on how to build efficient classifiers with little labeled data. Active learning technology provides a theoretical basis for future research on the identification of pine wilt diseased trees. Next, we will conduct research on tasks such as the classification of diseased trees in the field of active learning.

## 5. Conclusions

In this paper, to solve the problem of the poor detection effect of existing target detection algorithms on small-sized PWDT, we propose a new target detection network, SPW-FEN, for the detection of PWDT. First, to solve the problem that the shallow feature layer in the existing detection algorithms has insufficient ability to extract the features of small-sized diseased trees, in this paper, a PWCA attention module is proposed and adds the module to the shallow feature map, effectively improving the algorithm's ability to extract the features of small-scale diseased trees. Moreover, because of the problem that there are too few small-sized diseased trees in a single image, we propose an STE data enhancement method which effectively increases the number of small-sized diseased trees in a single image. The method proposed in this paper can effectively enhance the feature extraction ability of the network for small-sized diseased trees, reduce the missed detection rate of small-sized diseased trees, and achieve efficient detection of small-sized diseased trees in UAV images under complex backgrounds. The experimental results show that the method proposed in this paper has a recognition average precision of 79.1% and a recognition recall of 86.9% for pine wilt diseased trees. The recall and average precision are 3.6% and 3.8% higher than the current state-of-the-art method, Faster-RCNN [22]. At the same time, they are 6.4% and 5.5% higher than those in the YOLOv6 [28] algorithm in the latest YOLO series network.

In the future, we will focus on studying how to improve the detection performance of late-stage diseased trees, and use semi supervised feature learning and detection methods on the basis of a small amount of data samples to construct low-cost and high-precision diseased tree detection models. Additionally, we will further study the effect of mixed trees on the identification results of diseased trees, and verify the method for the possibility of error due to the presence of trees of other species (mixed forest).

**Author Contributions:** Conceptualization and methodology: S.Y., Y.Z. and H.S.; Software and validation: M.Y. and H.S.; Formal analysis: Y.J., C.H. and Y.S.; Investigation: Y.Z.; Resources: M.Y.; Data curation: H.S.; Writing—original draft preparation: S.Y. and Y.P.; Writing—review and editing: M.Y. and Y.Z.; Visualization: Y.P.; Supervision: H.S.; Project administration: Y.Z.; Funding acquisition: M.Y., Y.Z. and H.S. All authors have read and agreed to the published version of the manuscript.

**Funding:** This research was partially supported by the Natural Science Foundation of Hubei Province of China (Grant No. 2021CFB004).

**Informed Consent Statement:** Not applicable.

**Data Availability Statement:** Data are available upon reasonable request from the corresponding author.

**Conflicts of Interest:** The authors declare no conflict of interest.

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
