# Peer review of "A Shallow Pooled Weighted Feature Enhancement Network for Small-Sized Pine Wilt Diseased Tree Detection"

_electronics, doi:10.3390/electronics12112463_

Round 1

Reviewer 1 Report

The reviewed article discusses the results of testing new method to detect small-size pine wilt diseased trees. A Shallow Pooled Weighted Feature Enhancement Network based on Small Target Expansion has been proposed for solving the problem of the poor detection effect of existing target detection algorithms on small-size targets. The article is very interesting, the method is decribed precisely and in detail, the presented data are novel and the results are very promising.

Perhaps it would be better if Authors could add information about targets sizes (lines 137-140) not only in pixels but also in actual size of the tree (especially minimum size of the small tree). Also it would be good in future to verify the method for the possibility of error due to the presence of trees of other species (mixed forest) (perhaps it could be mentioned in conclusions?).

Reviewer 2 Report

The authors proposed a method for detecting Pine Wilt Diseased Trees (PWDTs)  using a Shallow Pooled Weighted Feature Enhancement Network (SPWFEN) based on Small Target Expansion (STE). However, there are a few concerns to be addressed:

1. In the abstract, the lines from 18 to 23 are to be rephrased in order to discuss the additional methods used in their work and obtained overall accuracies compared with the existing state of art methods. 

2. check thoroughly the entire manuscript for improper text format and the figure descriptions. 

3. A paragraph description of Small Target Extension (STE) for PWDTs detection should be presented in the introduction with the relevant references.

4. Figures should be labeled with a,b, c, and d if they contain 3 or 4 sub-images.

5. And also the description of the figures also should follow a consistent format explaining the sub-images i.e., a,b,c,d in a figure.

6. Section 3.3 is missing the proper heading.

7. The authors need to specify or indicate the hyper-tunning parameters used for their approach clearly in the corresponding tables and in the discussion as well.

8.  In line 375, "Tabular 7" was mentioned by the authors which is not appropriate and needed to be corrected.

9. The authors need to check for all these grammatical mistakes in the entire manuscript through thorough manuscript reading.

10. Finally the authors need to indicate/ describe their results in detail in the conclusion to improve the reader's interest. 

The authors need to check the grammatical errors in headings, figure descriptions, and tables,  thoroughly in the entire manuscript.  

Reviewer 3 Report

The paper is very interesting to read and well organized. The authors clearly present and synthetize the aim of this study, data set and data collection are discussed in detail. The results are discussed and presented in appropriate form. An all-round good paper, with some formatting and content changes required. In particular:

line 46, line 50, etc. - references are not in accordance with the guidelines of the journal (commas and spaces between square brackets are missing, it would be more appropriate to use e.g. [4-6] instead of [4], [5], [6], I think...)

line 53 - there is a Capital letter used in the text, i.e. "Xu Xinluo et al. [12] Used ...".

Fig. 5. (instead of Figure 5.) - Please, could you improve figure 5? In some countries, a comma is used instead of a decimal point, so I would change the text "521, 10%" to "521 (10%)", for example.

An inappropriately used capital letter in the middle of a sentence/or an inappropriately worded sentence, e.g. line 372.

Reviewer 4 Report

Dear Authors,

The manuscript entitled “A Shallow Pooled Weighted Feature Enhancement Network for Small-Size Pine Wilt Diseased Tree Detection” deals with an interesting topic that fits perfectly in the journal’s scope. However, it has some issues that have to be handled.

Concerning the major issues, it is not clear how diseased trees were identified exactly (i.e., manually, one by one, who did the identification, how could you state if a diseased-looking tree suffered from pine wilt disease and not from another disease, etc.?). It is also not clear what “Test Set” means in Table 2, while in the paragraph above only training and verification (and not “Validation” as in Table 2) data sets were explained/mentioned. Consequently, the composition of Table 2 is needed to be explained. Besides, abbreviations have to be explicated when first appearing. 

The spelling and wording of the manuscript have to be developed, since there are numerous typos and unclear sentences in it (revise, e.g., the sentences in lines 81-84, 123-124, 298-302, and 372-373). In sum, it is recommended to get the manuscript proofread by a professional, native English speaking  proofreader.

Round 2

Reviewer 2 Report

By reviewing the revised manuscript, it's  been observed that  the authors have addressed all the concerns from the previous review.

By reviewing the revised manuscript, It's been observed that the authors have corrected most of the grammatical errors.